# Mechanism of Action of Ketogenic Diet Treatment: Impact of Decanoic Acid and Beta—Hydroxybutyrate on Sirtuins and Energy Metabolism in Hippocampal Murine Neurons

**DOI:** 10.3390/nu12082379

**Published:** 2020-08-08

**Authors:** Partha Dabke, Anibh M. Das

**Affiliations:** Clinic for Pediatric Kidney, Liver and Metabolic Diseases, Hannover Medical School, 30625 Hannover, Germany; dabke.partha@mh-hannover.de

**Keywords:** beta-hydroxybutyrate, decanoic acid, energy metabolism, ketogenic diet, mitochondria, sirtuins

## Abstract

The ketogenic diet (KD), a high-lipid and low-carbohydrate diet, has been used in the treatment of epilepsy, neurodegenerative disorders, inborn errors of metabolism and cancer; however, the exact mechanism/s of its therapeutic effect is not completely known. We hypothesized that sirtuins (SIRT)—a group of seven NAD-dependent enzymes and important regulators of energy metabolism may be altered under KD treatment. HT22 hippocampal murine neurons were incubated with two important KD metabolites–beta-hydroxybutyrate (BHB) (the predominant ketone body) and decanoic acid (C10), both accumulating under KD. Enzyme activity, protein, and gene expressions of SIRT 1-4, enzyme capacities of the mitochondrial respiratory chain complexes (MRC), citrate synthase (CS) and gene expression of monocarboxylate transporters were measured in control (untreated) and KD-treated cells. Incubation with both–BHB and C10 resulted in significant elevation of SIRT1 enzyme activity and an overall upregulation of the MRC. C10 incubation showed prominent increases in maximal activities of complexes I + III and complex IV of the MRC and ratios of their activities to that of CS, pointing towards a more efficient functioning of the mitochondria in C10-treated cells.

## 1. Introduction

Ketogenic Diet: The term “Ketogenic Diet” (KD) was introduced by Dr. Wilder at the Mayo clinic in the early 1920s, wherein a ketonemia-producing diet was designed to treat patients with epilepsy. This diet was widely used for several years, but with the introduction of diphenylhydantoin in 1938, the research focus shifted from a ketogenic diet to pharmacological antiepileptic compounds [1]. A novel diet producing an equally effective ketonemia, termed as the “Medium Chain Triglyceride” (MCT) diet was successfully used by Dr. Huttenlocher in Chicago to treat children with pharmacoresistant epilepsy [2,3]. KD is a high-fat, low-carbohydrate diet used in the treatment of pharmaco-refractory epilepsy [4]. Various different types of KD have been proposed over the last few decades including the classical KD, modified Atkins diet (MAD) and the MCT diet [5]. Despite the differences in compositions, several studies have proven the therapeutic effect of all these different dietary regimes [6]. Besides epilepsy, KD is used as a therapeutic option in mitochondrial disorders such as Complex I deficiency [7], glycogen storage disease type 3 [8], neurodegenerative disorders such as Parkinson’s disease and Alzheimer’s disease [9], as well as cancer [10].

The exact mechanism of action of KD in epilepsy still remains elusive [11]. Although multiple biochemical and molecular pathways have been suggested to play a role in exerting the therapeutic effect of KD, beta-hydroxybutyrate (ßHB), the predominant ketone body, has been proposed to play an important role [12,13]. In addition to ketonemia, raised plasma levels of decanoic (C10) and octanoic acids (C8) are found under a ketogenic diet [14]. Especially, decanoic acid has been suggested to contribute to the mechanism of action in KD. Previous studies have shown that treatment with decanoic acid leads to improved mitochondrial function and energy metabolism in fibroblasts from patients affected with mitochondrial disorders [7] as well as in neuronal cell models [15]. Understanding mitochondrial metabolism and pathways that influence the mitochondrial function is crucial as mitochondrial biogenesis, improved energy metabolism and reduced mitochondrial ROS-levels have been hypothesized to be involved in the antiepileptic mechanism of KD [11,15,16,17].

Sirtuins: The term “Sirtuin” is derived from Sir2 protein (belonging to the silent information regulator family), a group of proteins described in Saccharomyces cerevisiae (yeast), particularly pertaining to its function in longevity [18]. Seven sirtuins (SIRT) have been described in mammals with diverse physiological functions [19]. These NAD^+^-dependent enzymes are ubiquitously present in the human body and have specific subcellular localizations, enzymatic activities and substrates [20]. Sirtuins have been shown to regulate crucial processes such as mitochondrial biogenesis, ROS detoxification and lipid metabolism; all of which presumably play an important role in KD (refer to Table 1). An alteration in energy metabolism, mitochondrial biogenesis and changes in glucose and lipid metabolism are amongst the important potential mechanisms of action in KD [11,12,17].

Mitochondrial respiratory chain (MRC): Mitochondria are the main site of energy production via the respiratory chain/electron transport chain (oxidative phosphorylation). Upregulated mitochondrial function and mitochondrial biogenesis have been postulated as one of the potential mechanisms of action in KD [11,16,17].

Changes in mitochondrial function may be based on altered sirtuin function. SIRT3 has been shown to regulate complex I [31], complex II [32,33], complex III [34,35], complex IV [36,37] as well as ATP–synthase (complex V) [27,38] of the mitochondrial respiratory chain. Furthermore, SIRT1 has been implicated in the regulation of mitochondrial biogenesis [21,39] and ROS detoxification [23,40]. This demonstrates a strong connection between mitochondrial energy metabolism and sirtuins. Citrate synthase (CS), an important enzyme of the Krebs cycle, often serves as a marker for mitochondrial content [41,42,43]. KD was shown to increase CS activity in cell culture [15] as well as in an animal model [44].

Monocarboxylate transporters: The monocarboxylate transporters (MCTr *) belong to a family of SLC16 solute carrier transporters [45]. They are responsible for transport of monocarboxylate molecules such as lactate and pyruvate as well as ketone bodies–beta-hydroxybutyrate, acetoacetate and acetate [46]. Although 14 members of this transporter family have been identified, function of only a few (i.e., MCTr 1–4, MCTr 8 and MCTr 10) has been well described [45,47]. MCTr 1 and MCTr 2 are expressed in varying amounts in different parts of the mouse brain, both these transporters are detectable in appreciable quantities in the cerebral cortex and the hippocampus [48,49,50]. These observations suggest that MCTrs may play an important role in the uptake of ketone bodies during KD treatment. Higher levels of MCTr 1 were observed in the brain endothelial cells of rats under KD treatment [51]. A recent study showed that MCTrs 1 and 2 play an important role in the uptake of ßHB during KD treatment in mice and inhibition of MCTrs in KD-treated animals results in an increase in epileptiform activity [52]. These observations suggest that MCTrs may play a crucial role in transportation of metabolites elevated under KD.

*The abbreviation ‘MCTr’ (as against the standard ‘MCT’) is used for ‘monocarboxylate transporter’ in this article since ‘MCT’ has been used for ‘medium chain triglyceride’.

**Hypothesis** **H1.** 
*We hypothesized that metabolites accumulating under KD, namely beta hydroxybutyrate and decanoic acid, alter sirtuin function, energy metabolism and MCTr expression in neuronal cells. Sirtuins may provide the vital link between KD and mitochondria as “regulators” of crucial biochemical processes such as mitochondrial biogenesis and energy metabolism, which have been implicated as potential mechanisms of the antiepileptic effect exerted by KD.*


## 2. Materials and Methods

Cell culture (for sirtuin analysis and MCTr): Hippocampal neurons from an immortalized murine cell line, HT22, were cultured in Dulbecco’s Modified Eagle’s Medium (DMEM) (Thermo Scientific, USA) with 10% (*v/v*) fetal bovine serum (Biowest SAS, France) and 1% (*v/v*) Penicillin/Streptomycin (Sigma-Aldrich, Steinheim, Germany). T25 sterile flasks (Sarstedt AG, Germany) were used to culture the cells. The cells were grown under 37 °C and 5% CO_2_. In the first phase of the study, HT22 neurons were cultured under two different glucose concentrations, standard (4.5 g/L) and low glucose (1 g/L). A quantity of 4.5 g/dL (450 mg/dL) of glucose seems high compared to human conditions but is the standard concentration for incubating HT22 cells. Concentrations lower than 1g/L (100 mg/dL) of glucose are not tolerated by HT22 neurons, proliferation is poor, as observed in pilot experiments. The low glucose cultures were established to simulate a “ketogenic” environment (low carb) for the cells. In the second phase of the study, cells were divided into either control (untreated) or KD-treated groups. Both the control and the KD-treated cells were cultured under low (1 g/L) glucose.

Two KD metabolites were used to treat the HT22 cells: 5 mM ßHB (beta–hydroxybutyrate), the predominant ketone body and 250 µM C10 (decanoic acid), a medium-chain fatty acid. This concentration of C10 is found in CSF of mice undergoing KD-treatment while blood concentrations (410 µM) are considerably higher [53]. Five millimoles ßHB is the typical therapeutic concentration in plasma reached in human patients under KD. In rats, a close positive correlation between blood- and CSF- concentrations of ßHB has been observed [54]. Ketonemia has been shown to increase the uptake of ketones by the brain [55]. A recent review on KD stated, that 5 mM of ßHB is a clinically meaningful concentration which is typically used for experimentation regarding KD [13]. Cells were incubated with decanoic acid or ßHB for 1 week with medium changes every other day.

Cells were lysed using Laemmli buffer for Western blot analysis, HEPES buffer for sirtuin enzyme activity and RLT buffer (Qiagen, Hilden, Germany) for RNA isolation.

Sirtuin enzyme activity: Enzyme activities of SIRT 1-3 were determined using the fluorescence-based enzyme activity kit (Enzo Life Sciences, Lausen, Switzerland), with SIRT1 (FLUOR DE LYS^®^ SIRT1 Substrate) (Enzo Life Sciences, Lausen, Switzerland) and SIRT2 and 3 (FLUOR DE LYS^®^ SIRT2 Substrate) (Enzo Life Sciences, Lausen, Switzerland) as specific substrates. A fluorescence plate reader (Tecan group Ltd., Männedorf, Switzerland) was used to detect the reaction.

Bicinchoninic assay (BCA): The BCA test was used to determine the protein concentration in the probes using the Pierce BCA protein assay kit (Thermo Scientific, Waltham, MA, USA). A full-area 96 well plate was used, and the colorimetric reaction was detected.

Protein expression: Equal number of cells (5000 cells/µL) were lysed using Laemmli buffer post harvesting. Semi-dry Western blotting technique was used to determine protein expression of sirtuins. SDS–PAGE gels (10% separating gel and 4% collecting gel) were used for electrophoresis and a nitrocellulose membrane was used for the transfer. A total of 10 µL of each probe was loaded onto the gel after heating at 95 °C for 15 min, followed by centrifugation at 10,000 rpm. Rabbit anti-SIRT1 (Millipore, Darmstadt, Germany), rabbit anti-SIRT2 (Santacruz, Heidelberg, Germany), rabbit-SIRT3 (Abcam, Cambridge, UK), goat anti-SIRT4 (Thermo Scientific, Waltham, MA, USA) and rabbit anti-Beta-Actin (Cell Signaling Technology, Inc., Danvers, MA, USA) primary antibodies were used in the recommended concentrations. Beta-Actin was used to normalize the protein expression of sirtuins in the respective probes. Relative protein expression was determined by normalizing with the control (untreated) probe. Secondary anti-rabbit and anti-goat antibodies were purchased from Li-cor Biosciences, USA. Intercept™ blocking buffer (TBS) from Li-Cor Biosciences was used to dilute all antibodies. The Odyssey FC system (Li-cor biosciences, Lincoln, NE, USA) was used for detection of bands.

Gene expression: RNA isolation and cDNA synthesis were performed using the RNeasy mini kit and Omniscript RT kit, respectively, from Qiagen, Germany. SIRT 1-4 and MCTrs 1 and 2 gene expression was analyzed using a SYBR green-based real-time polymerase chain reaction (PCR) on a 7900HT fast Real-Time PCR system and analyzed using SDS 2.4 software (Applied Biosystems, Foster, CA, USA). Murine primers for SIRT 1–4, MCTr 1, MCTr 2 and housekeeper genes (B2M, HPRT1 and ACT-B) were used for the PCR (refer to Appendix A. Beta-2-microglobulin (B2M), hypoxanthine phosphoribosyltransferase 1 (HPRT1) and beta-actin (ACT-B) were used as internal controls and relative sirtuin gene expression was calculated based on a normalization method using the three internal controls [56].

Cell culture (for MRC complexes): HT22-cells were grown in Nunclon™ Delta Surface petri dishes (Thermo Scientific, Waltham, MA, USA) at 37 °C and 5% CO_2_. Both control and KD-treated groups (either 5 mM ßHB or 250 µM C10) were grown under low glucose concentrations. The Petri dishes were washed using HEPES buffer followed by sonication (for 10 s x 2 times at 20 W power with single pulses of 0.3 s duration) using the HD 70 Sonopuls sonicator (BANDELIN electronic GmbH & Co. KG, Berlin, Germany) as described before for other cell types [57,58].

Spectrophotometric analysis of MRC complexes and CS: Capacities (maximal enzymatic activities under substrate saturation in vitro) of MRC complexes and Citrate Synthase (CS) were determined spectrophotometrically. Specific substances were used as inhibitors to measure activities of different complexes: rotenone (complex I + III), antimycin (complex II + III) and oligomycin (complex V). Spectrophotometric measurement was performed according to the established methods for complex I + III [59], complex II + III [60], complex V [61,62] and citrate synthase [63] at 37 °C. Precipitation of protein for quantification was done using trichloroacetic acid (TCA) [64] and subsequently, the protein concentration was measured using a BCA assay as described above.

Statistical analysis: All results are shown as mean + SD. The unpaired t-test (or Mann–Whitney test for non-parametric data) was used to determine the statistical significance of all the results. A *p*-value of less than 0.05 was considered as significant. A minimum of 3 independent samples were used for each experiment, measurements were carried out in duplicate. GraphPad Prism (San Diego, CA, USA) software version 8 was used for statistical analysis.

## 3. Results

### 3.1. Measurement of Sirtuins under Standard and Low Glucose Concentrations

The standard glucose concentration in DMEM used to culture HT22 neuronal cells is 4.5 g/L (25 mmol/L). To simulate a “ketogenic environment”, the glucose concentration was reduced to 1 g/L (5.6 mmol/L) in DMEM, the minimal concentration tolerated by the cells in terms of proliferation. The protein and gene expression were normalized to the low glucose (1 g/L) group. No statistically significant differences were noted between the two groups in enzyme activity (Figure 1A–C), relative protein expression (Figure 1D–F) and gene expression (Figure 1J–L) of sirtuins.

### 3.2. Incubation with the Predominant Ketone Body-BETA-Hydroxybutyrate (ßHB)

#### 3.2.1. Sirtuin Enzyme Activity, Protein Expression and Gene Expression

Both the untreated (control) and treated (with 5mM ßHB) groups of HT22 neurons were cultured under low (1 g/L) glucose in DMEM. ßHB incubation significantly induced SIRT1 enzyme activity (Figure 2A) as well as protein expression (Figure 2D) in comparison with the control. Similarly, enzyme activity of SIRT3 (Figure 2C) was also significantly increased on treatment with ßHB. The enzyme activities of SIRT1 and SIRT3 were increased ~2 fold and ~1.5 fold, respectively, as compared to controls. However, no significant change was seen in the protein expressions of SIRT2 (Figure 2E) and SIRT3 (Figure 2F), while protein expression of SIRT1 was significantly increased (Figure 2D). SIRT4 protein expression (Figure 2M), although not statistically significant, was ~1.4 fold higher in the ßHB-treated group as compared to the control. The consistent elevation in enzyme activities of SIRT 1–3, but no significant alterations at protein (except for SIRT1) and gene expression levels suggest a post-translational modification under 5 mM ßHB treatment.

#### 3.2.2. Mitochondrial Respiratory Chain (MRC) Complexes and Citrate Synthase (CS)

Enzyme activities of complex I+III, complex II+III, complex IV, complex V (ATP synthase) and the mitochondrial marker enzyme, citrate synthase (CS) were measured in control (untreated) and 5 mM ßHB-treated HT22 cells. Activities of complexes were normalized to CS as a mitochondrial marker. Incubation of HT22 neurons with 5 mM ßHB resulted in an upregulation of the mitochondrial respiratory chain. Enzyme activity of Complex I + III (Figure 3A) was significantly increased in the treatment group. Increases in the enzyme activities of Complex II + III (*p* = 0.0558) (Figure 3B) and Complex V (*p* = 0.0667) (Figure 3G) were non-significant. This suggests an overall induction of the respiratory chain complexes. Enzyme activity of citrate synthase (Figure 3I) was also significantly elevated in the ßHB-treated cells. CS has been described as a marker for mitochondrial content and an induction in its activity could point towards increased mitochondrial biogenesis.

### 3.3. Incubation with the Medium Chain Fatty Acid Decanoic Acid (C10)

#### 3.3.1. Sirtuin Enzyme Activity, Protein Expression and Gene Expression

Only SIRT1 enzyme activity (Figure 4A) was significantly elevated on C10 incubation, as compared to the untreated cells. SIRT3 enzyme activity (Figure 4C) showed a near-significant increase (*p* = 0.0581) in cells incubated under C10. Relative protein and gene expressions of SIRT3 were significantly higher in C10-incubated cells (Figure 4F,L). Albeit not significantly, SIRT4 was ~1.5 fold higher in the C10-incubated cells as compared to the control group (Figure 4M). A similar trend was also noted in the gene expression of SIRT3 (Figure 4L). Thus, decanoic acid mainly led to elevated SIRT1 enzyme capacity without a significant change in its protein or gene expression, once again suggesting a post-translational modification.

#### 3.3.2. Mitochondrial Respiratory Chain (MRC) Complexes and Citrate Synthase (CS)

Enzymatic capacities of complexes I–V as well as capacities normalized to the citrate synthase are reported. Decanoic acid (C10) incubation significantly increased the enzyme capacities of complex I + III (Figure 5A) and complex IV (Figure 5C). The activity of complex II + III (Figure 5B) showed a slight increase in the treatment group; however, this change was not significant. Interestingly, the enzymatic activities of ATP synthase (complex V) (Figure 5G) and citrate synthase (CS) (Figure 5I) remained unchanged in the C10-treated group. The ratios of maximal activity of the respiratory chain enzymes to that of CS demonstrate the enzyme activity of the respective complex in a given amount of mitochondria. It is noteworthy that the ratios of complexes I + III (Figure 5D) and complex IV (Figure 5F) normalized to CS in the C10-treated group were significantly higher than in the control group. This observation points towards a more efficient functioning of the mitochondrial respiratory chain in the C10 treated HT22 cells rather than an increase of mitochondrial mass.

### 3.4. Monocarboxylate Transporters (MCTr) 1 and 2

Gene expressions of MCTr 1 and MCTr 2 were determined in HT22 cells incubated with 5 mM ßHB and 250 µM C10. Significant upregulation of MCTr 1 in cells incubated with 250 µM C10 (Figure 6C) and MCTr 2 in cells incubated with 5 mM ßHB (Figure 6B) was observed. Non–significant upregulations of MCTr 1 in ßHB-treated cells (Figure 6A) and MCTr 2 in C10-treated cells (Figure 6D) were seen.

## 4. Discussion

Clinically, KD has been successfully used as a therapeutic option in epilepsy [1,2], other neurological disorders [5,9,65], inborn errors of metabolism [66] as well as in cancer [10]; however, the mechanism underlying these clinical benefits is still unclear. Mitochondrial dysfunction has been shown to be involved in the pathogenesis of epilepsy [67,68], neurodegenerative disorders [69,70], inborn errors of metabolism [58,71,72] and cancer [73]. Upregulation of the mitochondrial function/mitochondrial biogenesis has been suggested as a possible mechanism of action of KD [11,16,74].

Role of SIRT3: Sirtuins regulate mitochondrial metabolism as well as other biochemical and physiological pathways. About 65% of the mitochondrial proteins are acylated [75], offering a huge potential for sirtuins acting as NAD^+^-dependent deacylase enzymes [20], thus regulating mitochondrial function. Indeed, SIRT3 has been shown to deacetylate a wide variety of mitochondrial targets [34,76,77]. SIRT3 was found to regulate the mitochondrial respiratory chain complexes. A recent study described the role of SIRT3 in regulating mitochondrial biogenesis and bioenergetics in the context of KD treatment [17]. Long-chain acyl coenzyme A dehydrogenase (LCAD) was shown to be deacetylated and activated by SIRT3 in mice [26]. This shows that the fatty acid oxidation in mitochondria is regulated by SIRT3, an important aspect with respect to high-fat KD. In a recent in vivo- study, we have shown that exercise as a metabolic stressor leads to an up-regulation of sirtuins in humans [78]. Our results show an induction in enzyme activity of SIRT3 on incubation with 5 mM ßHB, the predominant ketone body. Incubation with decanoic acid showed an upregulation of SIRT3 protein and gene expression, which may be an additional mechanism increasing SIRT3 function in KD. NAD is increased under KD [79], which could additionally activate SIRT3 (and other sirtuins). SIRT3 function may be modulated by deacetylation or phosphorylation [80]; however, the mechanism of such a post-translational modification (PTM) in the context of KD is unclear.

Role of SIRT1: SIRT1 is predominantly localized in the nucleus and has a strong deacetylase activity [20]. Interestingly, it has also been suggested that SIRT1 may be found in the mitochondria as well; however, this is not well studied [39]. It has been shown that SIRT1, along with PGC-1α, regulates mitochondrial biogenesis [21,81]. Furthermore, SIRT1 has also been shown to regulate antioxidant mechanisms via the transcription factors PGC-1α and FOXO3a [22]. Studies have also reported that SIRT1 plays an important role in ROS detoxification via regulation of enzymes such as SOD2 and catalase [23]. Mitochondria are an important site for generation of ROS [82], thus making regulatory mechanisms of ROS detoxification by SIRT1 important in this context. Both ßHB and C10 treatment induced SIRT1 enzyme capacity in HT22 cells as compared to untreated cells. In response to incubation with 5 mM ßHB, SIRT1 protein expression was also increased. Enzyme capacity of citrate synthase as a marker for mitochondrial mass, was also significantly higher in the ßHB-treated HT22 cells indicating an induction of mitochondrial biogenesis. Taken together, these results support the hypothesis that improvement in mitochondrial function, via SIRT1, could be one of the potential mechanisms underlying the beneficial impact of ketogenic diet. Several different mechanisms of PTMs of SIRT1 have been described, including, but not limited to, phosphorylation, ubiquitination, and methylation [80,83]. A new PTM in the presence of high concentrations of the ketone body–ßHB has been described called lysine beta-hydroxybutyrylation [84]. Although this PTM has not been directly described in the context of sirtuins, it may provide valuable insights into the function of sirtuins in mediating the therapeutic effect of KD.

Upregulation of energy metabolism: Treatment of HT22 cells with both, ßHB and C10 resulted in an upregulation of the mitochondrial respiratory chain. The effect of C10 on enzymatic capacities of the MRC complexes in our study was prominent—an induction in not only the maximal activities of complex I + III and complex IV, but also their ratios normalized to the CS activity, indicating a more efficient function of the mitochondria, rather than an increase in mitochondrial mass. Our findings regarding upregulation of energy metabolism by C10, are in line with findings in previous studies, in the SH-SY5Y neuronal cell line as well as fibroblasts from patients with mitochondrial disorders [7,15]. However, the enzyme activity of CS was unchanged with 250 µM C10. On the other hand, treatment of HT22 cells with 5 mM ßHB induced activity of CS—suggesting a probable increase in mitochondrial quantity and function.

As described earlier, SIRT3 has been shown to regulate the function of the MRC. This is in line with our present study, where ßHB incubation of HT22 neurons significantly upregulated SIRT3 and respiratory chain enzyme activities. This can result in an overall upregulation of the respiratory chain. Mitochondrial biogenesis is an important potential mechanism of action of KD [16]. A recent article has described the role of SIRT3 in improving mitochondrial function and biogenesis on KD treatment [17]. These observations are similar to our analysis of sirtuins and energy metabolism on KD treatment in neuronal cells in vitro.

Upregulation of monocarboxylate transporters (MCTr): Levels of ßHB, the predominant ketone body, are significantly elevated during KD treatment and ßHB is taken up by the brain as an alternative energy substrate [12,85]. It is well established that MCTr 1 and 2 both facilitate transport of monocarboxylates such as ßHB in various tissues, including neurons [47,48,49]. Incubation of HT22 neurons with 5 mM ßHB resulted in significant upregulation of MCTr 2–indicating a possible role of the MCTr 2 in transporting ßHB present in the culture medium into the neurons. It has been shown that C10 is transported into the brain from systemic circulation; however, the mechanism behind this process in not known [53,65]. Our results demonstrate significant upregulation of gene expression of MCTr 1 in HT22 neurons incubated with 250 µM C10.

Findings of our study are summarized in Figure 7.

Clear limitations of our study are the in vitro model used which may not reflect in vivo conditions and the incubation with exogenous metabolites. Concentrations of ßHB vary in plasma and CSF under KD treatment, we used 5 mM, however there are individual differences in response to ßHB,-some patients even respond to lower concentrations. Further studies will be performed using lower serial concentrations (0.5 mM–2.5 mM) of ßHB to examine whether sirtuins and MRC complexes are differently altered under incubation with different ßHB concentrations. Under in vivo conditions, KD leads to an increase of both ßHB and decanoic acid which may have a synergistic effect.

## 5. Conclusions

The findings from our study indicate that two metabolites accumulating under ketogenic diet treatment, namely beta-hydroxybutyrate, the predominant ketone body, and decanoic acid—an important medium-chain fatty acid, induce the enzymatic activities of sirtuins and upregulate the mitochondrial respiratory chain enzymes in murine hippocampal neurons. Sirtuin up-regulation may result in the activation of respiratory chain enzymes. These observations suggest that sirtuins may play an important role in the mechanism of action of ketogenic diet, possibly via upregulated energy metabolism. However, further in vivo experimentation is needed to confirm this mechanism. In particular, the effect of inhibitors and activators of sirtuins on respiratory chain complexes shall be studied in future experiments.

## Figures and Tables

**Figure 1 nutrients-12-02379-f001:**
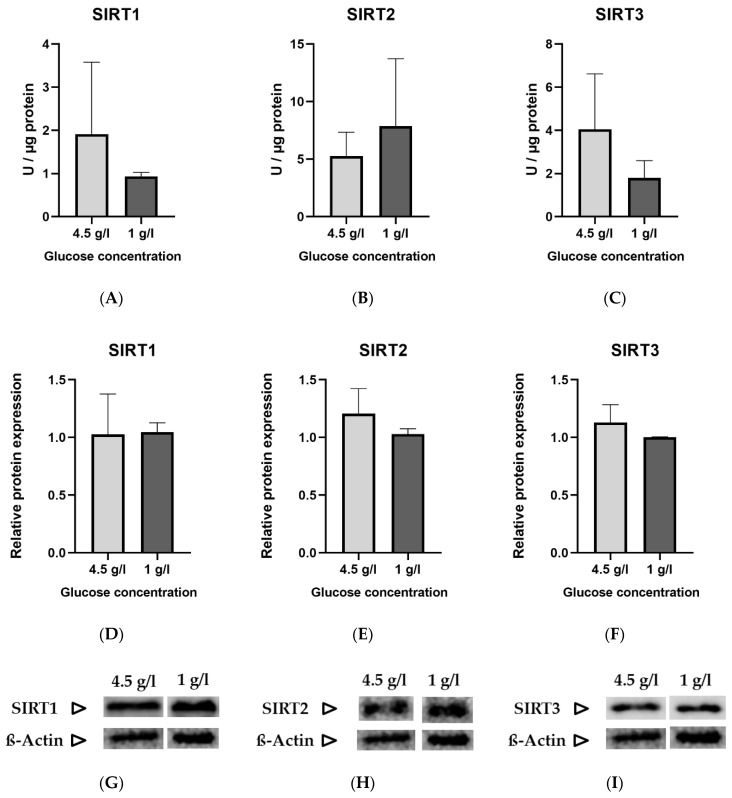
Graphs (**A**–**C**) represent absolute enzyme activities of SIRT 1–3 (Units/µg)., (**D**–**F)** represent relative protein expression, (**G**–**I**) are representative images of Western blots and (**J**–**L**) represent relative gene expression of SIRT 1–3. Data are expressed as mean + SD (*n* = 3–4).

**Figure 2 nutrients-12-02379-f002:**
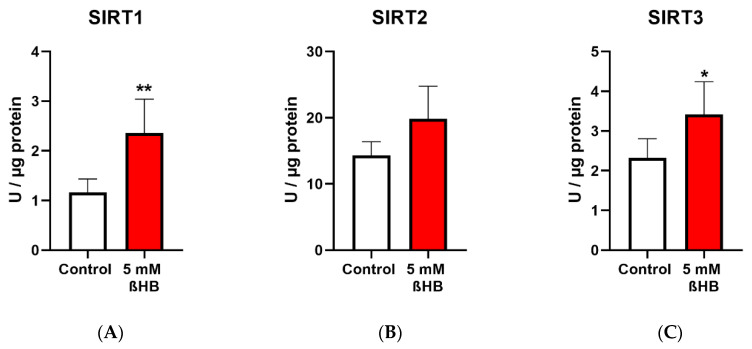
Graphs (**A**–**C**) represent absolute SIRT 1–3 enzyme activity (Units/µg), (**D**–**F,M**) represent relative protein expression and (**J**–**L**) represent relative gene expression of SIRT 1–4. (**G**–**I**,N,**O**) are representative images of Western blots. * *p* < 0.05, ** *p* < 0.01; (*n* = 3–8). Data are represented as mean + SD.

**Figure 3 nutrients-12-02379-f003:**
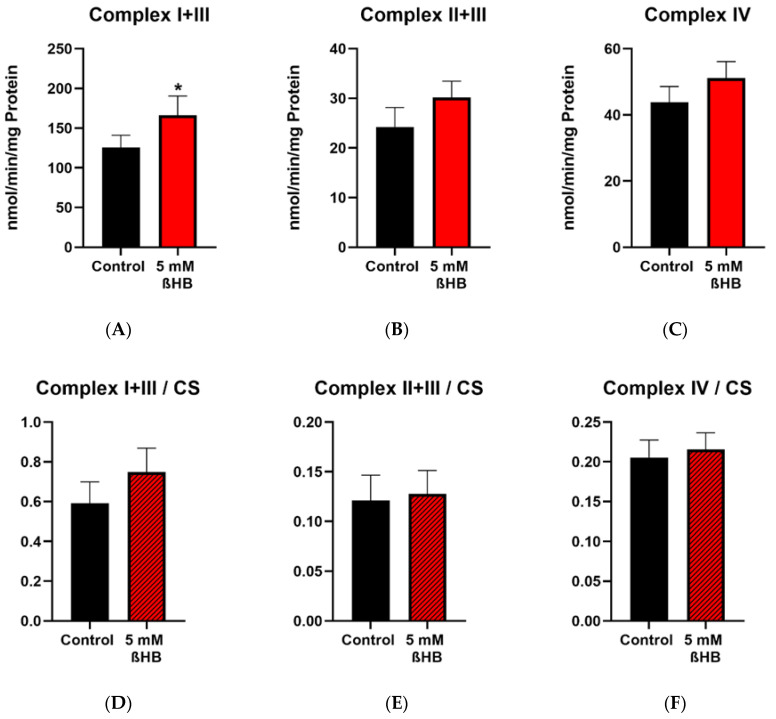
Graphs (**A**–**C**,**G**,**I**) represent maximal enzymatic activities of the respiratory chain complexes I–IV, ATP–synthase (complex V) and citrate synthase. The graphs (**D**–**F**,**H**) depict normalized ratios of the enzymatic activities of the respiratory chain to the enzymatic activity of citrate synthase–the mitochondrial marker enzyme. * *p* < 0.05 (*n* = 3–4). Data are shown as mean + SD.

**Figure 4 nutrients-12-02379-f004:**
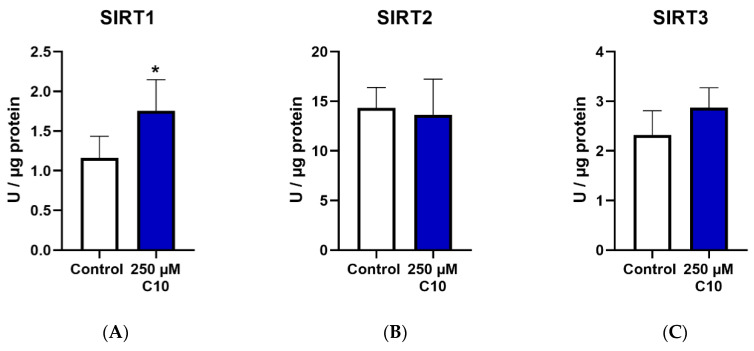
Graphs (**A**–**C**) represent absolute SIRT 1–3 enzyme activity., (**D**–**F**,**M**) represent relative protein expression and (**J**–**L**,**N**) represent relative gene expression of SIRT 1–4. (**G**–**I**,**O**) are representative western blot images. * *p* < 0.05 (*n* = 3–6). Data are represented as mean + SD.

**Figure 5 nutrients-12-02379-f005:**
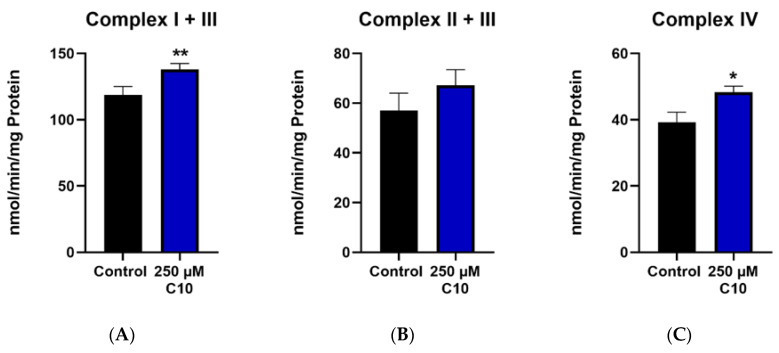
Graphs (**A**–**C**,**G**,**I**) represent maximal enzymatic activities of the respiratory chain complexes I–IV, ATP–synthase (complex V) and citrate synthase (CS). The graphs (**D**–**F**,**H**) depict ratios of the enzymatic activities of the respiratory chain to the enzymatic activity of citrate synthase–the mitochondrial marker enzyme. * *p* < 0.05; ** *p* < 0.01 (*n* = 3–4). Data are represented as mean + SD.

**Figure 6 nutrients-12-02379-f006:**
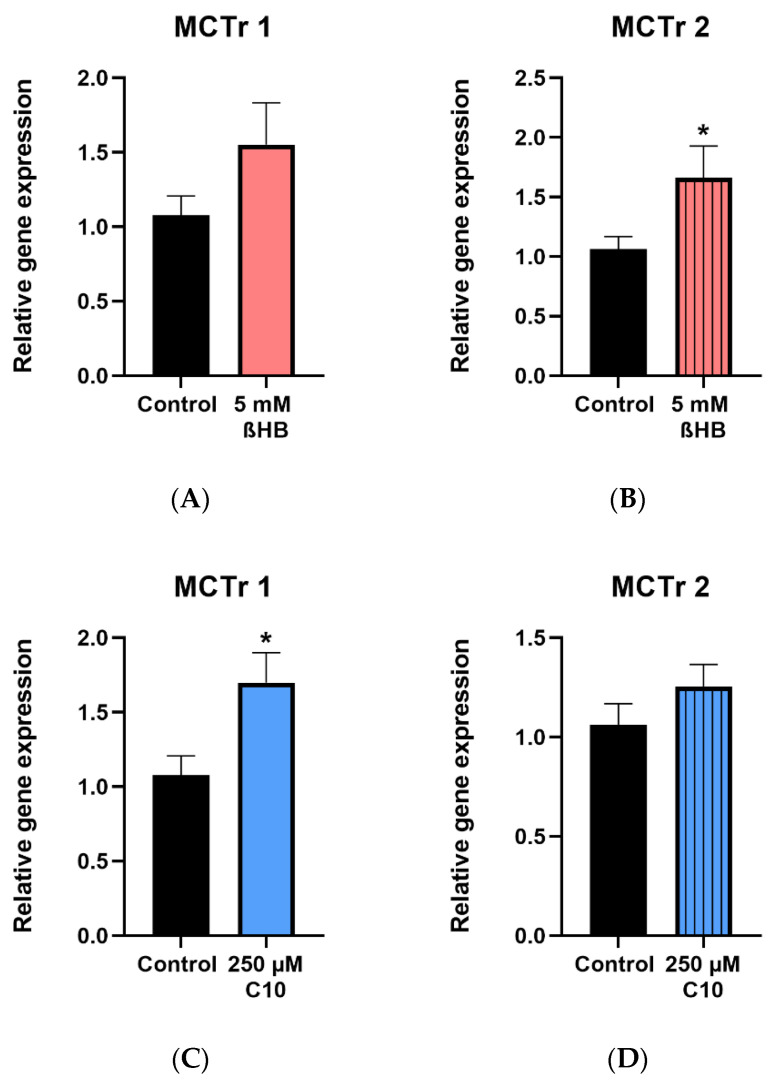
Gene expression of monocarboxylate transporters (MCTr 1 and MCTr 2) in 5 mM ßHB (**A**,**B**) or 250 µM C10-treated HT22 cells (**C**,**D**), relative to control (untreated) probes. Data are represented as mean + SD; * *p* < 0.05 (*n* = 3).

**Figure 7 nutrients-12-02379-f007:**
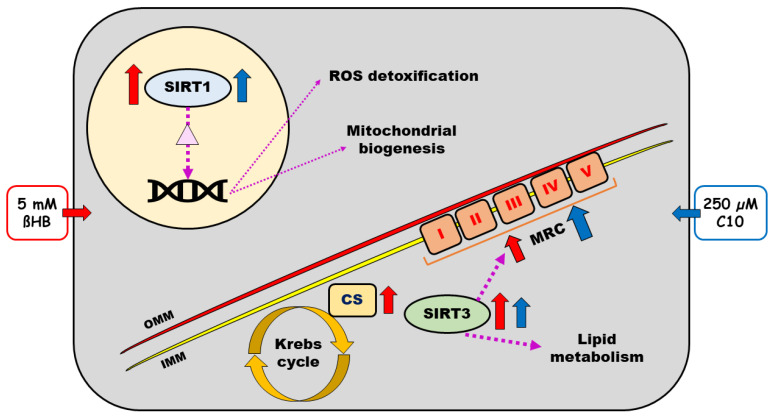
Schematic representation of a neuronal cell with sirtuins (Sirtuins 1–4) and respiratory chain and effects of beta-hydroxybutyrate (ßHB) on the same. Purple dotted arrows represent the known regulatory effects of SIRT 1–4. Solid red and blue arrows represent the effects of 5 mM ßHB and 250 µM C10, respectively. The light-yellow circle represents the nucleus with DNA and the light purple triangle therein represents transcription factors deacetylated by SIRT1 (PGC-1α and FOXO3a). ßHB—beta hydroxybutyrate; C10—decanoic acid; CS—citrate synthase; IMM—inner mitochondrial membrane; OMM—outer mitochondrial membrane; MRC—mitochondrial respiratory chain; ROS—reactive oxygen species; SIRT—sirtuin.

**Table 1 nutrients-12-02379-t001:** Sirtuins (SIRT 1–4): localization, enzymatic activity, * targets (relevant to potential mechanisms of action of ketogenic diet).

Sirtuin	Localization and Enzymatic Activity	Relevant Physiological Processes and Targets *	Reference
**SIRT1**	**Nucleus** *Deacetylation*	*Mitochondrial biogenesis* PGC-1α PPAR-γ	[21,22,23,24,25]
		*ROS detoxification* PGC-1α FOXO3a UCP-2	
		*Glucose metabolism* AMPK UCP2 (regulation of insulin secretion)	
**SIRT2**	**Cytoplasm** *Deacetylation*	*ROS detoxification* FOXO3a NF-ΚB	[23]
**SIRT3**	**Mitochondria** *Deacetylation*	*Energy metabolism (MRC)* (Discussed in detail in the next sub-section)	[26,27]
		*Fatty acid oxidation* LCAD Acetyl CoA synthetase 2	
**SIRT4**	**Mitochondria** *ADP-ribosylation* *(predominant)* *Deacetylation **(weak)***	*Fatty acid oxidation* Malonyl CoA decarboxylase PPAR-α MCAD PDK4 CPT1α PPAR-δ	[28,29,30]

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
