# Peer review of "Mechanism of Action of Ketogenic Diet Treatment: Impact of Decanoic Acid and Beta—Hydroxybutyrate on Sirtuins and Energy Metabolism in Hippocampal Murine Neurons"

_nutrients, 2020, doi:10.3390/nu12082379_

Round 1

Reviewer 1 Report

In the manuscript “Mechanism of Action of Ketogenic Diet Treatment:  Impact of Decanoic Acid and Beta-Hydroxybutyrate on Sirtuins and Energy Metabolism in Hippocampal Murine Neurons” the authors describe several in vitro studies that examine the impact of the ketogenic byproducts on sirtuins and mitochondrial activity in an effort to explore potential mechanisms that contribute to the beneficial effects of the ketogenic diet (KD).  Overall, the paper presents useful information that adds to the field, and the introduction and discussion are very clear.  However, clarification and additional information need to be provided in the methods and results.  The following changes are suggested:

  1. Did the media in which the cells were grown contain sodium pyruvate? As this is an alternative source of energy, it would be helpful to know if it is present in the media.
  2. The concentration of C10 is based on plasma levels in patients undergoing KD treatment, which may be higher than the C10 levels reached in the brain. Have brain C10 levels been explored?  If so, these levels should be used as opposed to plasma levels.
  3. While line 113 states “ . . . 5mM BHB is the typical therapeutic concentration reached under KD”, there is no reference, and is unclear whether this refers to blood or brain levels. This concentration seems very high as blood baseline levels are between 0.1 and 0.250 mM (Wildenhoff et al., 1974) and levels above 1-3 mM are indicative of ketoacidosis.  If the authors do not have a reference for this concentration of BHB in the brain, additional experiments should be completed with clinically relevant concentrations of BHB. 
  4. How long were cells treated with BHB and C10 in the various experiments?
  5. It is unclear why some experimental data was analyzed with paired t-tests while others used unpaired t-tests. Based on the written methods, it seems as if all data should be analyzed using unpaired t-tests, unless enzyme activity was measured before and after BHB/C10 treatment.
  6. In all figures that examined SIRT levels, the control group is set to a value of 1. While this is not problematic for relative protein and gene expression, enzyme activity should be reported in enzymatic units per mg protein.  Based on the written methods, it seems that the BCA assay was used to determine protein concentration for this set of experiments, but this should be made more clear with more details on how this was performed for different experiments (enzyme activities v. western blots).  Furthermore, even if control values are normalized to 1, error bars should portray standard deviation in the control group.
  7. Representative images of western blots, including loading controls, should be provided for all figures showing protein expression.

Author Response

Reviewer reports – Nutrients Manuscript

Dear editorial team,

First of all, we like to thank both reviewers and the editorial team for their constructive and helpful suggestions. We have made some alterations to the manuscript according to these suggestions.

Reviewer 1 report (round 1):

In the manuscript “Mechanism of Action of Ketogenic Diet Treatment: Impact of Decanoic Acid and Beta-Hydroxybutyrate on Sirtuins and Energy Metabolism in Hippocampal Murine Neurons” the authors describe several in vitro studies that examine the impact of the ketogenic byproducts on sirtuins and mitochondrial activity in an effort to explore potential mechanisms that contribute to the beneficial effects of the ketogenic diet (KD).  Overall, the paper presents useful information that adds to the field, and the introduction and discussion are very clear.  However, clarification and additional information need to be provided in the methods and results.  The following changes are suggested:

Did the media in which the cells were grown contain sodium pyruvate? As this is an alternative source of energy, it would be helpful to know if it is present in the media.

Ans.: The media we used did not contain sodium pyruvate.

The concentration of C10 is based on plasma levels in patients undergoing KD treatment, which may be higher than the C10 levels reached in the brain. Have brain C10 levels been explored?  If so, these levels should be used as opposed to plasma levels.

Ans.: A study by Wlaz et al (2015) reported plasma and brain levels of C10 in mice as 410 µM and 240 µM, respectively. Ref.: Wlaz et al (2015), Acute anticonvulsant effects of capric acid in seizure tests in mice (DOI: 10.1016/j.pnpbp.2014.10.013).

While line 113 states “ . . . 5mM BHB is the typical therapeutic concentration reached under KD”, there is no reference, and is unclear whether this refers to blood or brain levels. This concentration seems very high as blood baseline levels are between 0.1 and 0.250 mM (Wildenhoff et al., 1974) and levels above 1-3 mM are indicative of ketoacidosis.  If the authors do not have a reference for this concentration of BHB in the brain, additional experiments should be completed with clinically relevant concentrations of BHB.

Ans.: 5 mM of ßHB is a clinically meaningful CSF-concentration, this has now been specified along with respective references.

How long were cells treated with BHB and C10 in the various experiments?

Ans.: The cells were treated for 1 week with BHB and C10 in all the experiments.

It is unclear why some experimental data was analyzed with paired t-tests while others used unpaired t-tests. Based on the written methods, it seems as if all data should be analyzed using unpaired t-tests, unless enzyme activity was measured before and after BHB/C10 treatment.

Ans.: The data have now been analyzed using the unpaired t-test (or Mann Whitney test) throughout in the revised version.

In all figures that examined SIRT levels, the control group is set to a value of 1. While this is not problematic for relative protein and gene expression, enzyme activity should be reported in enzymatic units per mg protein.  Based on the written methods, it seems that the BCA assay was used to determine protein concentration for this set of experiments, but this should be made more clear with more details on how this was performed for different experiments (enzyme activities v. western blots).  Furthermore, even if control values are normalized to 1, error bars should portray standard deviation in the control group.

Ans.: Absolute enzyme activities of SIRT1 – 3 have been provided in the revised version. The new graphs portray standard deviation bars. The cell pellet was collected in HEPES buffer for enzyme activity and the same cell lysate was used to measure the protein concentration. For western blotting, equal number of cells (5,000 cells per µl) were lysed using Laemmli buffer and 10 µl of each probe was loaded into the gel. Beta-Actin was used for normalization for each probe.

Representative images of western blots, including loading controls, should be provided for all figures showing protein expression.

Ans.: Representative images have been provided in the revised version of the manuscript.

Reviewer 2 Report

In this manuscript, Dabke P and Das A explored the mechanism of action of a ketogenic diet in murine neurons by analyzing the effect of decanoic acid (C10) and Beta – Hydroxybutyrate 3 on Sirtuins expression and activity, mitochondria biogenesis and activity. This study is relevant considering the prescription of the ketogenic diet in the treatment of epilepsy and other neurological disorders. Overall, the manuscript is clearly written and organized.

There are some points that the authors should consider:

  • The western-blotting results for analysis of Sirtuins expression are not shown, only the quantification of the blots. They should be included in the manuscript.
  • Lines 181-185- This paragraph is not easily understood. What do authors mean by untreated cells, when they were comparing low with high glucose effect on cells?
  • Citrate synthase activity is usually used as mitochondria biogenesis marker. However, considering that enzyme activity was elevated in Beta – Hydroxybutyrate 3 treated cell, authors should perform additional experiments to confirm the effect of this ketogenic product on mitochondria biogenesis and exclude an effect on enzyme activity. For example, mitochondria content should be evaluated with a fluorescent labelling or by electron microscopy.
  • Figure 7- only statistically significant results should be presented in the figure. For example, Sirt4 expression is not increased by Beta – Hydroxybutyrate 3 or C10 treatment, as indicated in the figure.
  • Line 323- authors conclude an increase of SIRT3 activity on incubation with 5 mM ßHB. Since activity of sirt3 was measured in addition to sirt2 it is not possible to conclude this.

Minor points:

-Lines 291-292- Sentences need revision.

Line 353- “regarding”- typo

Round 2

Reviewer 1 Report

In this revision, the authors have provided significantly improved graphs and have modified the statistical analysis to be more appropriate to the experimental design.  However, the text describing the references provided to support the 5mM BHB concentration needs to be improved.  For example, the authors state that Wang et al. shows a positive correlation between serum and plasma BHB levels.  While accurate, this is not indicative that the levels are the same; the CSF levels are 69% of the serum levels.  In fact, the paper shows that 2 weeks of a KD in rats leads to a CSF concentration of only 65 uM, much lower than the 5mM used in the paper.  While there are several studies that use concentrations of 5mM, the review article referenced (Simeone et al.) specifies clinically relevant BHB levels to be 0.05 - 2.5mM.  The paper would be significantly strengthened with the use of lower concentrations of BHB; at the very least, this limitation should be specified in the discussion section.

Author Response

Dear Reviewer,

Thank you for your suggestions. We have now added this limitation of BHB concentration, in the discussion section as recommended.

Thank you

Kind Regards